# Giving Voices: Qualitative Study on Parental Experiences of Caring for Children with Cerebral Palsy or Developmental Disabilities in South Korea

**DOI:** 10.3390/children12030284

**Published:** 2025-02-26

**Authors:** Bogcheon Choi

**Affiliations:** Department of Rehabilitation, Jeonju University, Jeonju 55069, Republic of Korea; splbc@jj.ac.kr

**Keywords:** children with disabilities, parenting challenges, parental lived experiences, family support systems, qualitative research

## Abstract

**Background/Objectives**: This study investigates the lived experiences of Korean parents caring for children with cerebral palsy or developmental disabilities, focusing on the challenges they face. It highlights systemic and societal factors, including gaps in diagnostic processes, financial support, access to information, and inclusive education. **Methods**: Using a qualitative approach, semi-structured interviews were conducted with 17 parents, recruited through purposive sampling. Thematic analysis was employed to identify key patterns and challenges in their experiences. **Results**: Parents reported significant difficulties, including delayed and insensitive diagnostic processes, financial burdens due to inadequate welfare support, and limited access to coordinated information. Educational challenges included disabling attitudes among teachers and a lack of genuine inclusion. Societal stigma further compounded their struggles. Despite these obstacles, parents demonstrated resilience and advocated for their children’s needs. Despite these obstacles, parents demonstrated resilience and advocated for their children’s needs. **Conclusions**: This study provides valuable insights into the challenges faced by parents of disabled children in South Korea, emphasizing the need for systemic reforms to foster a more inclusive and supportive environment for these families.

## 1. Introduction

The concept of disability has evolved over the past several decades, influenced by social, historical, and ideological contexts. Traditionally, academic research and policy discourse adhered to the medical model, which views disability through the lens of chronic illness, positioning individuals within the framework of a ‘sick role’ [1]. This perspective assumes that impairments inherently hinder participation in normal activities, framing disabled individuals as incapable of fulfilling conventional social roles. Consequently, their disadvantages and exclusions have been framed as personal problems caused by their impairments. The solutions within this framework typically focus on “curing” the bodily condition or encouraging individuals to “adjust” to their circumstances [2]. A closely related concept is the personal tragedy model, which views impairment as making individuals “less than whole” [3]. This paradigm requires disabled individuals to rehabilitate their impairments and attempt to achieve lower-valued social roles [4]. These models reinforce the idea that disability is an individual problem, neglecting broader structural and societal factors that contribute to exclusion and inequality [2,5].

Medical and individualistic approaches to disability have long been critiqued by disabled theorists and advocates, who call for a comprehensive re-evaluation of disability issues. A pivotal development in this discourse is the social model of disability [6,7]. This model challenges biological determinism and essentialist perspectives by distinguishing between impairment and disability [4]. Within this framework, disability is understood not as an inherent characteristic of the individual but as the result of social structures and processes. The difficulties and limitations experienced by disabled individuals are seen as arising not from their impairments but from societal barriers and power imbalances [5,8].

The debate between the medical and social models of disability has not only shaped discussions on disabled individuals’ rights but has also influenced research, policy, and services for families raising disabled children [9,10]. Ferguson [11], in his review of post-World War II literature, argues that traditional family research largely extends the medical model. This perspective frames parental challenges as direct consequences of the child’s impairment, with family needs viewed as requiring therapeutic interventions targeting the child’s condition [10,11]. Moreover, much of the early research on families with disabled children often viewed their experiences through a medical or deficit-based perspective, assuming a direct causal link between the presence of a disabled child and family dysfunction. This perspective often characterized families as experiencing maladjustment, with an absence of a “normal” family dynamic [12,13]. Such portrayals contributed to biased research, where the impact of disability on parents’ well-being and family life were mostly shown in a negative light [14,15]. These studies frequently highlighted cases of parents perceived as struggling to manage the demands of raising a disabled child, reinforcing deficit-based narratives [12,16].

Since the 1990s, scholars have increasingly challenged the deficit-based narratives surrounding disability and parenting. While acknowledging the social pressures parents face, research has also highlighted the positive dimensions of raising a disabled child, such as enhanced family harmony, strengthened cohesion, spiritual growth, and a deeper empathy for others’ experiences [17,18]. Rather than viewing their disabled children as burdens, many parents perceive their relationships as reciprocal and mutually enriching [19,20]. Scholars have also emphasized the importance of environmental factors in shaping disability experiences. An emerging body of literature calls for a re-evaluation of previously pathologized aspects of disability by considering the influence of environmental contingencies [21,22]. As Read [23] observes, the challenges of raising a disabled child can be viewed in two main ways: first, as consequences of the child’s impairment and its impact on parental well-being, and second, as the result of disabling societal barriers embedded within the caregiving experience. Increasing acknowledgment is being given to the importance of understanding the experiences of disabled children and their families within the context of larger social, cultural, and political frameworks [24,25].

Building on this shift toward understanding disability within broader societal contexts, scholars have sought theoretical frameworks to better capture the interplay between individuals and structural influences. One such framework is Bronfenbrenner’s ecological systems theory (EST), which serves as an analytical tool for examining how various societal and institutional structures shape parental experiences [26,27,28]. EST posits that human development is influenced by interactions across multiple environmental systems, including the microsystem (family, school, healthcare institutions), mesosystem (relationships between institutions, such as school–home interactions), exosystem (policy and welfare systems), and macrosystem (disability stigma and cultural attitudes) [26].

By integrating Bronfenbrenner’s EST with the social model of disability, this study seeks to explore the lived experiences of disablement and the perspectives of parents raising disabled children in South Korea. While extensive research exists on Korean families with disabled children, much of it remains grounded in psychological and therapeutic models that emphasize parental attitudes and behavioral adjustments [12]. Consequently, these studies often frame family challenges as issues of maladjustment, leading to individualized interventions rather than addressing structural barriers. In contrast, this study reinterprets the challenges faced by parents of disabled children through the lens of disability rights and social justice, highlighting the social, cultural, and material conditions influencing their lived experiences [25]. Using qualitative research methods, this study delves deeply into parents’ lived realities to highlight their voices and provide a more nuanced understanding of disablement. The remainder of this paper is organized as follows: Section 2 describes the methodology, Section 3 presents the findings, and Section 4 discusses the results and their policy implications.

## 2. Materials and Methods

### 2.1. Research Design and Participants 

This study employed a qualitative exploratory design to examine the parenting challenges of families raising disabled children. These challenges were identified as the central phenomenon requiring in-depth exploration, aligning with the qualitative research objective of capturing participants’ lived experiences and perspectives within their social realities. This study aimed to provide a comprehensive understanding of how parents experience and interpret these challenges in the context of raising a disabled child.

Participants were recruited through purposive sampling in two ways: (1) engaging with four parents’ self-help groups in Seoul, South Korea, to invite interested participants, and (2) employing a snowball sampling technique, where participants referred others with similar experiences. To ensure sufficient depth of data, this study aimed for thematic saturation, where new data no longer yielded novel themes. Saturation was assessed by continuously analyzing the transcripts during data collection [29]. By the 15th interview, no significantly new themes emerged, and the last two interviews confirmed existing themes, indicating saturation. While the sample size may not allow for broad generalization, it was deemed sufficient for capturing rich, in-depth insights within this qualitative framework.

Table 1 provides demographic details of the participants. All parents came from middle- or upper-middle-class households, as defined by income levels. Their ages ranged from 38 to 52, with the majority being in their forties. All families consisted of two parents, with thirteen mothers and four fathers. According to the parents, the children’s conditions included intellectual disabilities, cerebral palsy, autism, and Down syndrome.

While participants shared a relatively homogeneous socioeconomic background, this study aimed to develop an in-depth contextual understanding of parents’ experiences, offering rich insights into their lived realities.

### 2.2. Data Collection

The interviews were conducted between April 2024 and September 2024. Data were collected through semi-structured interviews, which allowed for an in-depth exploration of parenting experiences, social barriers, and support needs. While interviews followed a predefined set of questions, they were iteratively adapted to incorporate emerging themes, ensuring a dynamic and participant-driven exploration of experiences [30]. The initial questions focused on key parenting challenges, including (RQ1) “Can you describe your experiences of raising a disabled child?”, (RQ2) “What challenges have you encountered in this context?”, (RQ3) “How do these challenges affect you and your family?” and (RQ4) “What types of support do you believe parents need when raising a disabled child?”.

As data collection progressed, the interview protocol was refined in response to emerging themes, ensuring that the study captured nuanced and evolving aspects of participants’ lived experiences [30]. For example, early interviews revealed that many parents were concerned about how their children were treated at school, including challenges in receiving adequate support. As a result, follow-up questions in later interviews evolved to explore teachers’ attitudes, relationships between parents and teachers, and parents’ experiences advocating for their child’s educational needs. While the core structure remained consistent across participants, this adaptive approach allowed for a more organic and participant-driven dialogue, enhancing the study’s credibility. Parents were encouraged to share personal stories and critical life events, ensuring that their perspectives on disablement were authentically represented.

All ethical considerations were addressed in accordance with the Declaration of Helsinki [31]. Prior to the interviews, participants were provided with an information letter outlining the purpose of the study and confirming that participation was voluntary. They were also informed about the confidentiality of their responses and their right to withdraw from the study at any time. Written informed consent was obtained from all participants before the interviews commenced.

### 2.3. Data Analysis

An inductive thematic analysis approach was employed to identify patterns and themes within the data. All interviews were transcribed verbatim to ensure accuracy. The data analysis process followed the six recursive phases outlined by Braun and Clarke [32], summarized as follows: (1) Becoming familiar with the data: reading the data multiple times and noting initial ideas. (2) Generating initial codes: systematically coding meaningful features and collating relevant data for each code. (3) Searching for themes: organizing codes into potential themes. (4) Reviewing themes: examining the themes for coherence with coded extracts and their relevance to the entire dataset. (5) Defining and naming themes: refining each theme, providing clear definitions, and assigning descriptive names. (6) Writing up findings: selecting illustrative extracts and contextualizing the analysis within the research questions and existing literature.

While the primary analysis was conducted by a single researcher, measures were implemented to enhance the credibility and trustworthiness of the findings. First, to minimize the risk of bias, a reference group consisting of two external experts (one PhD researcher and one university professor) reviewed the coding framework and thematic structure. Their feedback was incorporated to ensure the themes accurately represented the data. Second, a member-checking process was conducted, in which selected participants reviewed the interpreted findings to confirm that the analysis appropriately reflected their experiences [29]. These procedures strengthened the validity of the analytical process and ensured that the themes were firmly grounded in the data.

## 3. Results

Through a detailed analysis of the transcripts, three overarching master themes were identified. These themes, along with their corresponding sub-themes, are summarized in Table 2.

### 3.1. Facing Early Struggles of Parenting a Disabled Child

This central theme highlights the journey of parents as they adjust to raising a child with a disability, emphasizing the unique challenges they encountered during the early years. Since these events took place in the past for all participants, their reflections were influenced as time progressed and their lived experiences evolved. However, the subsequent years offered a broader perspective, enabling them to articulate the barriers and challenges they had faced with greater clarity. Key challenges identified by the parents included understanding their child’s differences, dealing with disclosure and professionals, accessing information, and coping with negative reactions from relatives.

#### 3.1.1. Recognizing Differences and Seeking a Diagnosis

Parents’ initial recognition of their child’s differences was often influenced by the nature of the impairment. Among the participants in this study, only four children exhibited clear signs of impairment at birth. As illustrated below, when impairments were not immediately visible, most parents initially viewed their child’s differences as typical developmental variations, often attributing them to being slower to develop:


*“I thought something was a bit off, but everyone around me, especially other moms, kept saying, ‘Kids are different. Boys can walk later.’ So, I tried not to worry. But as time passed, I got more uneasy. I started thinking—what if something’s actually wrong?”*
(Participant #4)

As these differences persisted, parents could no longer ignore their suspicions and sought a medical diagnosis for their child. However, the pursuit of a diagnosis did not always result in one. Some mothers reported that medical professionals frequently dismissed their concerns, often viewing them as overly sensitive, as illustrated below:


*“I told the doctor about my baby’s seizures, but he brushed it off, saying, ‘That’s normal for infants. You’re overreacting.’ Another doctor even snapped, ‘Are you trying to label your child as abnormal?’ It wasn’t until the fourth doctor that someone finally believed me—my baby had a seizure right in front of him.”*
(Participant #8)

When a clear diagnosis was delayed, parents faced prolonged periods of uncertainty and anxiety, often consulting multiple doctors in search of an accurate diagnosis for their child’s condition. One mother recounted:


*“For nearly a year, I kept going from one hospital to another. I saw doctor after doctor, hoping for answers. It cost me a lot—both time and money.”*
(Participant #13)

This period of “diagnostic limbo”, marked by suspicion and a lack of confirmation, was highly stressful [12]. Most parents expressed that early awareness, and diagnostic information would have been invaluable. Although they did not view a diagnosis as a life-long sentence, they felt it was necessary to move forward and plan effectively for their child’s needs.

#### 3.1.2. Disclosure and Disabling Professionals’ Attitudes

Many parents in this research described emotionally distressing experiences in diagnostic settings, primarily due to medical professionals’ failure to acknowledge the value and individuality of their child. As illustrated by one mother’s account, medical professionals often dismissed her child’s potential and portrayed their future as bleak, overlooking the child’s unique personhood:


*“The doctor’s words really stung. He didn’t seem to care how I felt. He said something like, ‘Give him a block, and he’ll be fine playing alone all day. Who needs friends?’ As his mom, that broke my heart. I couldn’t believe a doctor would say that.”*
(Participant #5)

Parents also reported that medical professionals often presented a more negative outlook on the future prospects for their child than expected. A mother recalled feeling deeply hurt by a doctor’s dismissive remarks during the diagnostic process:


*“The doctor glanced at me and said I was still young. He suggested I could have another baby, a healthy one. He made it sound like this child was a lost cause. I left feeling crushed. How could he speak so lightly about giving up? Why didn’t he help me see a way forward?”*
(Participant #16)

As these narratives illustrate, the disclosure of a diagnosis often framed the disabled child as ’incapable of being normal’. This framing did not simply describe the child’s impairment as a deviation from typical developmental milestones but conveyed a broader message of hopelessness and meaninglessness in the child’s life. Such messages placed an extra emotional strain on parents, who were already grappling with the reality of their child’s disability, further intensifying their anxiety and frustration.

#### 3.1.3. Struggles with Information Access

Parents in this study identified the lack of access to crucial information and guidance as one of the greatest challenges during the early years. They struggled to find information about services that could address both their child’s needs and their own support requirements. One mother reflected on her experience:


*“I didn’t know what to do or where to turn. I had no clue who could help or what kind of treatment my child might need. Early on, I felt stuck and overwhelmed.”*
(Participant #11)

The lack of access to information left many parents feeling anxious, helpless, and frustrated as they tried to navigate unfamiliar systems without adequate guidance. In the absence of formal resources, many parents turned to personal and informal networks, making great efforts to gather vital information. This often led to exhaustion and significant stress. One mother described the overwhelming burden of gathering information:


*“Everything was on me. I ran around on my own, trying to find information. It felt like searching for water in a desert. In the end, I was exhausted.”*
(Participant #8)

As a result, parents suggested that access to comprehensive resources providing practical information on raising a disabled child, along with clear guidance on available support services, would be highly beneficial. They emphasized the importance of establishing a coordinated support system to ensure that future parents of disabled children can access timely and accurate information, alleviating the emotional and practical burdens they experienced.

#### 3.1.4. Dealing with Relatives’ Negative Attitudes

Relatives’ attitudes play a particularly significant role during the early adaptation period. Assistance from relatives, including emotional support and genuine acceptance of the disabled child, represents a crucial resource for parents during this demanding time [33]. However, some parents in this study experienced considerable stress due to negative reactions from relatives toward their disabled child. In some instances, grandparents regarded having a disabled child in the family as a “source of shame”. One mother shared her experience:


*“My son’s grandfather didn’t like him going outside. He wanted to keep his disability a secret. He seemed worried that people would judge him if they found out.”*
(Participant #7)

Another significant aspect of Korean Confucian culture revealed in this study was the tendency to blame mothers, as having a disabled child was often linked to a defect attributed to the mother’s family line. Parents who faced such reactions often tried to resolve the conflict but, in some cases, had to sever family ties to protect both their child and themselves. One father shared his story:


*“After my son’s disability was known, my mom and sister wouldn’t stop calling. They pushed for separation. I was breaking down. One day, I told them, ‘She’s my wife, and we’re raising our son together. Don’t contact me again.’ We didn’t speak for three years after that.”*
(Participant #2)

### 3.2. Navigating the Challenges of Daily Family Life with a Disabled Child

This master theme captures the challenges parents face in managing their daily family life with a disabled child. Despite differences in their circumstances, parents commonly experienced financial strain, caregiving burdens due to limited support, emotional stress from ongoing worry about the future, and difficulties meeting the needs of non-disabled siblings.

#### 3.2.1. Financial Burden

Families raising a disabled child are often at increased risk of financial hardship. As widely recognized, the intensive care demands of a disabled child frequently restrict parents, particularly mothers, from fully participating in the labor market [34]. This loss of earning capacity was consistently described by mothers in this study as a major obstacle to ensuring their family’s financial stability.


*“Losing my ability to earn money put our family in a tough spot. Raising a child with special needs costs so much. The programs he attends are expensive. We don’t earn much, but spending for him never stops. It feels like we’re sinking deeper into poverty.”*
(Participant #3)

The most commonly cited and ongoing concern among parents was the financial burden of meeting their child’s specialized needs, including medical treatments, rehabilitative services, and assistive devices. In the absence of sufficient government support to ease these disability-related expenses, many parents were left to rely entirely on their personal financial resources.


*“We need a new specialized wheelchair for our child. A regular one could harm his body. But it’s too expensive … insurance won’t cover any of it … He also needs surgery later this year because his limbs are getting worse. We had no choice but to take out a bank loan.”*
(Participant #1)

Financial hardship stood out as one of the most pressing practical difficulties for parents, affecting their daily lives. In response, some families adopted coping strategies such as cutting household expenses, taking on extra work, or working overtime to better meet their child’s needs.


*“I often work night shifts, about 80 h a month. On Saturdays, I finish late in the evening, and sometimes I get a day off during the week. … It’s exhausting, but as a dad, I feel I have to … Without it, we couldn’t afford my child’s care.”*
(Participant #2)

#### 3.2.2. Lack of Social Support in Caregiving

Caring for a child with a disability is a central aspect of parents’ daily lives. As widely recognized, managing caregiving responsibilities often requires a combination of formal and informal support systems. Formal care services, such as daycare programs and respite care, can offer vital relief by reducing parental stress and improving family well-being [35]. However, for many families in this study, these formal care services were either inaccessible or severely limited.


*“The daycare program is limited to three years because there are so many on the waiting list. They’re really strict about it.”*
(Participant #14)


*“There was nowhere close by for my child to go. The nearest daycare for disabled children was an hour’s drive away.”*
(Participant #12)

Families raising children with severe disabilities faced even greater difficulties in accessing care services due to a shortage of appropriate service providers.


*“Daycare centers won’t take kids with severe disabilities like my son. They only accept those with mild conditions. If you go, you’ll see one worker handling four or five children. It’s just not set up for kids like ours.”*
(Participant #4)

This led some parents to perceive the support services as inadequate for their families. As a result, many parents were left to bear the full burden of caregiving for their disabled children alone.

#### 3.2.3. Emotional Burden of Worry and Unmet Needs

Parents demonstrated remarkable resilience in managing their family lives, but many also experienced persistent feelings of insecurity and vulnerability. They were acutely aware that intensive caregiving responsibilities would likely extend into their child’s adulthood, which led to ongoing stress and anxiety about their family’s future. Across the interviews, a recurring concern was the constant worry about financial stability as their child grew older. This stress was often exacerbated by additional challenges, such as the parents’ own health problems.


*“Raising a disabled child feels like pouring water on dry ground. We can never save anything. What if I get sick one day? How would my family cope? Thinking about it keeps me up at night.”*
(Participant #1)

Parents also faced emotional struggles in balancing the needs of their disabled child with those of other family members. They often had to sacrifice their own needs, as well as those of their non-disabled children, to prioritize the essential care requirements of their disabled child. This constant tension frequently led to feelings of guilt and undermined their parental sense of self-worth.


*“My other daughter hoped to start piano classes, but that would have meant cutting one of my son’s therapy sessions. It’s always a trade-off like that. I feel guilty all the time. I hate that I can’t give her more.”*
(Participant #8)

#### 3.2.4. Coping with Non-Disabled Sibling’s Challenges

The non-disabled siblings of disabled children often experience what some parents described as a “secondary form of disability” as they navigate the social stigma surrounding their disabled siblings. Negative societal perceptions of disability frequently extend to non-disabled siblings, resulting in their exclusion or negative treatment by peers [36]. This concern was repeatedly highlighted by parents during the interviews.


*“For the first two years of elementary school, my older daughter struggled with classmates picking on her. What hurt her most was being left out when others had birthday parties.”*
(Participant #6)

Due to the pervasive stigma surrounding disability, non-disabled siblings often internalized feelings of embarrassment about their disabled brother or sister [37]. These experiences led some to see their family as falling short of societal norms. One mother recounted how her child feared being judged as different because of their disabled sibling.


*“He was nine when he said something I’ll never forget. We were watching a TV drama about a man with a disability. All of a sudden, he said, ‘Mum, I want to marry a girl like his sister.’ I asked him why. He looked at me and said, ‘Because her family knows about kids like my brother. She won’t think I’m weird.’”*
(Participant #5)

Many parents felt the need to take on additional roles to support their non-disabled children in developing a positive understanding of their disabled sibling. Through daily conversations, they explained the nature of disability and made efforts to reduce the emotional strain caused by negative societal attitudes. Parents also sought to strengthen their non-disabled children’s self-esteem, as one mother described:


*“I’ve told her many times that families with disabled children can live happily too. I think she’s taken that to heart. She doesn’t see having a disability as something bad. She even tells her friends about her brother without any hesitation.”*
(Participant #6)

### 3.3. Parental Challenges in Schooling

This master theme captures parents’ experiences navigating their child’s education journey. Despite the diversity in educational environments, parents consistently encountered common challenges in securing appropriate support for their child’s education. They highlighted several specific obstacles, including teachers’ disabling attitudes, deficient collaboration between schools and families, and lack of access to appropriate educational opportunities.

#### 3.3.1. Teachers’ Disabling Attitudes

While some parents in this study recalled instances of positive attitudes and supportive behavior from certain teachers, many reported that their child was often treated unfavorably and set apart from their peers. Several mothers further identified teachers’ misconceptions about disability as a barrier to their child’s schooling and inclusion, as the following case illustrates:


*“My son is in a special class with four other kids who have mild disabilities. They never join the regular class or outdoor play. The head teacher keeps saying, ‘They could cause trouble or distract others.’ It feels like our kids are always seen as the problem.”*
(Participant #9)

Mothers further described how their children were often perceived as part of a ’dangerous group’. This perception stemmed more from teachers’ preconceived notions about disability than from the actual behaviors or characteristics of the children themselves. The following mother’s account vividly illustrates this point:


*“One day, the teacher called and asked me to come in. She said my son kept touching his pants near his private part during class. Right away, I knew—it must’ve been because he wet himself a little. I explained, but she didn’t believe me. What hurt most was when she bluntly asked, ‘Is he some kind of pervert?’”*
(Participant #10)

The parents’ accounts revealed that teachers’ negative attitudes and behaviors were closely linked to disabling cultural practices, including stigmatization and the categorization of disabled children as fundamentally different. The discriminatory values directed at the children often extended to their parents as well. Many parents reported feeling unfairly treated and receiving less respect and consideration than other parents.


*“We always feel judged just for being parents of children with disabilities. Some teachers don’t say thank you, even when we help at school events. It’s like they think we owe it to them because our kids make their jobs harder.”*
(Participant #6)

#### 3.3.2. Deficient Collaboration

Most parents felt it was essential to share their experiences and insights with teachers to ensure their child received the necessary support for school adjustment and educational progress. For instance, one father of a child with autism and behavioral challenges described his efforts to help teachers understand his child’s unique needs:


*“Most teachers lack experience teaching children with disabilities. They don’t know kids like my child. So for me, when a new homeroom teacher is appointed, I request a meeting to discuss my child. I provide the teacher with records, explain my child’s educational plan, why certain behaviors may occur, and how we should handle them.”*
(Participant #17)

However, many parents reported facing significant challenges in forming partnerships with teachers. A large number of them felt that their voices were dismissed or undervalued by some teachers, as one mother shared:


*“Last year, I was under a lot of stress due to my child’s homeroom teacher. As we all know, every child is unique, with their own traits. I wanted to share helpful tips and information about my child, but she didn’t seem to pay attention to anything I said.”*
(Participant #10)

Some parents mentioned that their participation in their child’s schooling was frequently seen as undermining the authority of teachers. One mother recalled how the teacher at a special school responded to her input:


*“Moms often feel that the curriculum doesn’t suit our child’s needs, and it should be adjusted. But when we try to offer suggestions, teachers usually aren’t receptive. A teacher once said to me, ‘If you think you know better, why don’t you handle it yourself?’”*
 (Participant #14)

Parents strongly believed that working together could lead to better solutions for their child’s school adjustment and educational development. However, parents experienced that their efforts to foster collaborative relationships with teachers were often disregarded because of a lack of respect and an unwillingness to address their concerns. This absence of partnership became a significant cause of frustration for parents, as they felt shut out from meaningful participation in their child’s education.

#### 3.3.3. Lack of Access to Educational Opportunities

A common concern expressed by parents regarding their child’s education was a shortage of adequate educational opportunities. Many parents who placed their children in special schools anticipated personalized and specialized educational services. However, their expectations frequently did not match the reality, as shown in the following account:


*“The school is called special, but the curriculum is just like any other school’s. We’ve asked several times for changes to better fit our children’s needs, but nothing has been done. What we’re asking for is real, individualized education for our kids.”*
(Participant #4)

Several parents expressed strong dissatisfaction with special schools, believing that these schools did not help their children achieve their full potential. One mother, feeling frustrated, compared the school to a childminding service:


*“A few students participate in class, while the others just sit there. Most of the time feels wasted. The teacher does everything alone. Honestly, the only benefit is that our child is looked after at school.”*
(Participant #6)

On the other hand, many parents who decided to send their children to mainstream schools after thoughtful consideration faced considerable stress due to insufficient support systems. Mainstream schools often failed to meet the individual needs of disabled students, causing adjustment difficulties. Parents commonly pointed to inaccessible environments, a lack of essential facilities and insufficient support staff as major obstacles. A mother of a child with mobility challenges shared her experience:


*“Our kids need help with everyday things like getting around, going to class, and using the restroom. There’s only one assistant for five kids, so sometimes they’re left alone. Some parents even stay at school to help their child.”*
(Participant #3)

Inclusive education was another major concern for parents. Alongside insufficient services, several parents pointed out that their children were often barred from taking part in typical school activities. Furthermore, mainstream schools frequently isolated disabled students in separate classrooms, which contradicted the principles of inclusive education. One mother emphasized:


*“Just because our children are in a regular school doesn’t mean they’re truly included. How can we call it an inclusive class when all the children with disabilities are taught in a separate room by just one teacher?”*
(Participant #9)

## 4. Discussion

This study explores the lived experiences and perspectives of Korean parents raising a disabled child, with a particular focus on the social and cultural contexts that shape their parenting challenges. The social model of disability provides a crucial framework for understanding systemic barriers. However, to provide a broader understanding, this study also incorporates Bronfenbrenner’s ecological systems theory, which offers a multi-layered perspective on how various environmental and institutional factors interact to influence parental experiences. This integrated approach not only addresses systemic and environmental challenges but also emphasizes policy implications that account for the complex realities these families face.

A key finding was the persistent issue of delayed diagnosis, which delayed access to essential interventions and increased parental anxiety and uncertainty, hindering early adaptation and family well-being. Many parents in this study reported the emotional toll of waiting for a definitive diagnosis. They acknowledged its necessity for accessing resources but found the process deeply stressful. These findings are consistent with prior research [38,39], which underscores the need for better diagnostic processes to facilitate early support. Policy interventions should focus on expanding early screening programs, improving medical training for diagnostic accuracy, and establishing clearer referral pathways between healthcare and education [40]. Interdisciplinary collaboration among pediatricians, educators, and social service providers is essential for early identification and timely intervention, ensuring that families receive appropriate support as early as possible [41].

In addition to diagnostic challenges, parents often experienced insensitive interactions with medical professionals during the disclosure process. Negative attitudes from professionals often conveyed messages of hopelessness, intensifying parents’ distress and making adjustment more challenging. Studies show that the manner of disclosure significantly influences how parents adapt to their child’s condition [42,43]. Policymakers should establish clear disclosure guidelines that prioritize empathy, cultural awareness, and clear communication [44]. Healthcare training programs must integrate disability awareness training to equip professionals for delivering sensitive and effective diagnostic disclosures [45].

Financial hardship also emerged as a dominant concern for parents. Although South Korea provides some welfare support, it often falls short of covering the costs of therapies, assistive devices, and educational support. This financial burden often resulted in compromises in other household needs, particularly as many mothers faced limited access to stable employment. The link between financial strain and parental stress is well established in disability research [46], highlighting the need for expanded financial assistance. Expanding financial support programs—such as increasing government subsidies for disability-related costs, broadening eligibility for cash benefits, and introducing tax incentives for families—could reduce financial strain and promote family stability [47].

Parents reported persistent difficulties in accessing accurate and timely information about services, benefits, and educational opportunities. Many relied on informal networks due to a lack of structured support to navigate complex welfare and education systems. The lack of coordinated information was particularly problematic during critical transitions, such as the initial diagnosis phase and entry into formal education. To address these gaps, a national platform should be developed to centralize disability-related resources and ensure families receive timely and accessible information [48]. In addition, dedicated family support coordinators across service sectors could offer families individualized guidance, particularly during key transition periods [49].

Consistent with prior studies [50,51], this research found that parents frequently expressed dissatisfaction with the limited availability and quality of services for disabled children. Parents reported difficulty obtaining tailored support, as existing services often failed to meet their children’s complex needs, particularly for those with severe or multiple disabilities. These findings underscore the pressing need for child-centered services that prioritize parental involvement, individualized support, and inter-professional collaboration to meet the complex and evolving needs of disabled children [52,53]. Services such as respite care, rehabilitation programs, and inclusive education opportunities can alleviate care burdens and enhance the overall quality of family life.

Beyond structural barriers, parents also encountered societal challenges, most notably stigma and social exclusion. Many mothers reported experiencing social isolation resulting from negative cultural perceptions of disability, which undermined their emotional wellbeing and strained their interactions with service providers. Public awareness campaigns and policy initiatives to promote inclusive attitudes and reduce stigma are vital for fostering a more supportive environment for disabled children and their families [54]. Disability awareness in schools, authentic media representation, and strengthened community inclusion efforts could foster acceptance and social integration [23].

Finally, the findings highlight ongoing limitations within the education system, a critical area for intervention. Although South Korea has made significant strides in special education, many parents reported that the rhetoric of inclusive education has yet to translate into reality [55]. Key barriers included untrained teachers, lack of collaboration between parents and educators, and exclusionary practices that limited disabled children’s participation in mainstream activities. These challenges underscore the need for comprehensive reforms, including mandatory special education training for teachers, expanding inclusive education resources in schools, and establishing structured parent–school partnerships to enhance communication and collaboration [56]. Ensuring that inclusive education policies are effectively implemented requires stronger governmental oversight and increased funding for specialized support programs.

Taken together, these findings underscore the necessity of adopting a multi-level policy approach that integrates family, institutional, and systemic factors. Despite these findings, this study is not without limitations. The sample primarily included families from metropolitan areas with relatively higher socioeconomic status, which may limit the generalizability of the findings. The effectiveness and accessibility of policies may also vary depending on families’ socioeconomic position and place of residence. Future research should examine how families’ access to and experiences with support systems differ across socioeconomic conditions and regional settings. Additionally, longitudinal studies could offer deeper insights into how parental experiences shift as their children grow and family needs evolve across different life stages.

Despite certain limitations, this study makes a valuable contribution to understanding the lived experiences of parents raising disabled children in South Korea. By integrating the social model with an ecological systems approach, it provides a more nuanced perspective on the structural, institutional, and cultural factors shaping parental experiences. Addressing these areas through evidence-based policies and cross-sector collaboration will be essential in ensuring that disabled children and their families receive the comprehensive support they require. Moving forward, concerted efforts from policymakers, educators, and service providers will be needed to build a more equitable and inclusive environment for all families navigating disability.

## 5. Conclusions

This study examined the lived experiences of Korean parents raising disabled children, highlighting the multifaceted challenges they face. Delayed diagnoses and insensitive communication from professionals exacerbated parents’ emotional distress, underscoring the need for comprehensive professional training and clear guidelines on disability disclosure. Financial hardships, stemming from the high cost of specialized care and limited welfare benefits, further strained families, revealing the urgent need for policies aimed at expanding financial support for these families. Barriers to accessing coordinated information and services, particularly during key transitional periods, continue to present significant obstacles. To alleviate these burdens, parents emphasized the importance of multi-agency approaches that streamline support. In schools, persistent issues such as teachers’ disabling attitudes and the lack of true inclusion reveal a gap between policy rhetoric and practical implementation, underscoring the need for reform in inclusive education policies. Lastly, societal stigma continues to marginalize families of disabled children, highlighting the need for public awareness campaigns aimed at reducing this stigma.

In conclusion, this study underscores the need for systemic changes to better support families of disabled children in South Korea. Integrating the social model of disability and ecological systems perspectives offers a comprehensive understanding of families’ realities and can inform policies that address financial, educational, and societal barriers. Developing family-centered policies through cross-sector collaboration will foster a more inclusive environment. Future research should explore the diverse experiences of families across different contexts and life stages to inform more targeted interventions.

## Figures and Tables

**Table 1 children-12-00284-t001:** Demographic details of the participants.

Participant	Parent	Child	Child’s Age (Years)	Child’s Disabilities
#1	Father	Son	8	Cerebral Palsy
#2	Father	Son	12	Cerebral Palsy
#3	Mother	Son	11	Cerebral Palsy
#4	Mother	Son	13	Cerebral Palsy
#5	Mother	Son	12	Autism
#6	Mother	Son	9	Autism
#7	Mother	Son	7	Intellectual Disability
#8	Mother	Son	9	Intellectual Disability
#9	Mother	Son	11	Autism
#10	Mother	Son	10	Autism
#11	Mother	Son	13	Intellectual Disability
#12	Father	Daughter	16	Intellectual Disability
#13	Mother	Son	17	Down Syndrome
#14	Mother	Son	15	Autism
#15	Mother	Daughter	14	Intellectual Disability
#16	Mother	Son	16	Down Syndrome
#17	Father	Son	9	Autism

**Table 2 children-12-00284-t002:** Master and sub-themes.

Master Theme	Sub-Theme
Facing early struggles of parenting a disabled child	- Recognizing differences and seeking a diagnosis - Disclosure and professionals’ disabling attitudes - Struggles with information access - Dealing with relatives’ negative attitudes
Navigating the challenges of daily family life with a disabled child	- Financial burden - Lack of social support in caregiving - Emotional burden of worry and unmet needs - Coping with non-disabled sibling’s challenges
Parental challenges in schooling	- Teachers’ disabling attitudes - Deficient collaboration - Lack of access to educational opportunities

## Data Availability

The data presented in this study are available on request from the corresponding author due to privacy and ethical restrictions.

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
