# Peer review of "Giving Voices: Qualitative Study on Parental Experiences of Caring for Children with Cerebral Palsy or Developmental Disabilities in South Korea"

_children, 2025, doi:10.3390/children12030284_

Round 1
Reviewer 1 Report
Comments and Suggestions for Authors
The manuscript presents an insightful qualitative study on the experiences of Korean parents raising children with cerebral palsy or developmental disabilities. The thematic analysis is thorough, but the discussion could engage more critically with disability theory beyond the medical and social models. Adding perspectives like Bronfenbrenner’s ecological systems theory could help contextualize findings. Methodologically, the sample size (n=17) lacks justification regarding saturation, and the socioeconomic homogeneity of participants limits generalizability. A more explicit discussion of these limitations would strengthen the study.
Additionally, while the interviews were semi-structured, the manuscript does not clarify whether questions evolved during data collection. Greater transparency here would enhance credibility. The findings on delayed diagnosis, financial burden, and educational challenges are compelling, but a stronger link to policy implications is needed. Overall, it’s a well-executed study that would benefit from deeper theoretical grounding and clearer methodological justification.
Reviewer 2 Report
Comments and Suggestions for Authors This paper is thoughtful and beautifully written. This paper explores the experiences of Korean parents caring for children developmental disabilities, starting with quite a deep dive into the models of disability and societal barriers. Through semi-structured interviews with parents of children with developmental disabilities, key themes emerged such as inadequate welfare support, societal stigma and discriminatory attitudes from others. Despite these obstacles, parents demonstrated resilience and were strong advocates for their children. The study urges systemic reforms to enhance support for these families in South Korea. There was a minor typo - Table 2 - "Mater & Sub-Themes" should be Master. Another weakness that could be mentioned is that when there is only one author coding and completing analysis there is a risk of bias.Author Response
"Please see the attachment."

Reviewer 3 Report
Comments and Suggestions for Authors
The paper has a broader social context, showing the numerous challenges faced by parents in raising children with various disabilities, which is a very important matter today. I suggest the following changes and additions:
1. At the end of the Introduction, present the structure of the paper in one paragraph.
2. State the year the research was conducted and in what period?
3. Table 1: (i) second column: instead of “gender”, “parent” should be used; (ii) the third column “Child (Age)” should be divided into two, so that the first is “child”, and the second is “child's age”.
4.Lines 127-130: the research questions are labeled with: “(1)”, “(2)”, “(3)” and “(4)”; instead, I suggest labels in the form: “(RQ1)”, “(RQ2)”, “(RQ3)” and “(RQ4)”.
5. Format Table 2 and remove bullets in the first column, while in the second use "–".
6. The discussion compares the results with research from 20-30 years ago, which is the main objection. I suggest that the comparison of the results be done with research of a more recent date. In general, many references in the paper are old, so where possible, they should be replaced with new – current ones.
Author Response
"Please see the attachment."

Round 2
Reviewer 3 Report
Comments and Suggestions for Authors
Accept in present form
Comments on the Quality of English LanguageAccept in present form